# A Lifelong Impact on Endometriosis: Pathophysiology and Pharmacological Treatment

**DOI:** 10.3390/ijms24087503

**Published:** 2023-04-19

**Authors:** Liang-Hsuan Chen, Wei-Che Lo, Hong-Yuan Huang, Hsien-Ming Wu

**Affiliations:** 1Department of Obstetrics and Gynecology, Linkou Medical Center, Chang Gung Memorial Hospital, College of Medicine, Chang Gung University, Taoyuan 33305, Taiwan; 2Graduate Institute of Clinical Medical Sciences, College of Medicine, Chang Gung University, Taoyuan 33302, Taiwan

**Keywords:** endometriosis, pathogenesis, inflammation, angiogenesis, medical therapy, pharmacological inhibitors

## Abstract

Endometriosis is a chronic inflammatory disease associated with bothersome symptoms in premenopausal women and is complicated with long-term systemic impacts in the post-menopausal stage. It is generally defined by the presence of endometrial-like tissue outside the uterine cavity, which causes menstrual disorders, chronic pelvic pain, and infertility. Endometriotic lesions can also spread and grow in extra-pelvic sites; the chronic inflammatory status can cause systemic effects, including metabolic disorder, immune dysregulation, and cardiovascular diseases. The uncertain etiologies of endometriosis and their diverse presentations limit the treatment efficacy. High recurrence risk and intolerable side effects result in poor compliance. Current studies for endometriosis have paid attention to the advances in hormonal, neurological, and immunological approaches to the pathophysiology and their potential pharmacological intervention. Here we provide an overview of the lifelong impacts of endometriosis and summarize the updated consensus on therapeutic strategies.

## 1. Introduction

Endometriosis is a common disease arising in adolescents that affects about 6–10% of women of reproductive age [1]. It progresses throughout menstrual cycles and involves multiple organs, resulting in local gynecologic lesions and systemic inflammatory disorders. Endometriosis is associated with a wide range of presentations, and its common symptoms include dysmenorrhea, dyspareunia, pelvic pain, dyschezia, and hematochezia. Furthermore, asymptomatic endometriotic lesions can be detected in nearly half of the women seeking infertility treatment [2]. As a condition of chronic inflammation and immune dysregulation, women with endometriosis are at higher risk of developing cardiovascular disease, rheumatoid arthritis, asthma, melanoma, ovarian cancer, and breast cancer [3].

Endometriosis is defined by the presence of endometrial-like tissue (‘‘lesions’’) outside the uterine cavity confirmed during surgery, where the diagnosis is often delayed after the onset of symptoms and mistaken because of its nonspecific complaints [1,4]. Nowadays, the diagnosis of endometriosis can be accelerated by advanced imaging techniques and associated serum biomarkers [5]. The conventional treatment includes surgical removal of endometriotic lesions followed by hormonal suppression. Current pharmacological treatments have limited efficacy and unwanted side effects. Half of the women undergoing surgery without long-term medication control may have another procedure in 5 years, resulting in organ damage complicated with loss of function [6]. Current therapeutic strategies highlight enduring symptom relief and fertility preservation [4,7]. The theories address whether the endometriotic cells travel to an abnormal location in a consequent mechanism or whether the endometriotic cells pre-exist in a milieu of genetic or epigenetic changes. A better understanding of the pathobiology of endometriosis in hormonal, inflammatory, metabolic, and pain pathways helps develop novel pharmacologic targets for clinical trials. In the present review, we provide an overview of the lifelong impact of endometriosis and summarize the current practice strategies in pathophysiologic and pharmacological aspects.

## 2. The Pathophysiology of Endometriosis

### 2.1. Retrograde Menstruation, Coelomic Metaplasia, and Müllerian Remnants Theory

Endometriosis is an “ectopic” endometriotic lesion that resembles the phenotype of the “eutopic” endometrial lining of the uterus. The most widely accepted hypothesis is retrograde menstruation that the efflux of menstrual tissue fragments via the fallopian tubes introduces endometrial cells seeding and growing into the peritoneal cavity [8]. Retrograde menstruation is a physiological event that occurs during menses, but in women with endometriosis, this viable steroid-responsive endometrial tissue adheres to the peritoneum and invades the pelvic structures. Higher risks for developing endometriosis in women with menarche at a younger age, menopause at an older age, long duration, and heavy menstrual flow support this hypothesis [9,10]. However, it is not sufficient to explain hereditary changes or clonal aspects of endometriosis in men, and its associated diverse clinical presentations.

Coelomic metaplasia, another well-recognized theory for forming endometriotic lesions, is based on the peritoneal mesothelium transformation. Moreover, the Müllerian remnants hypothesis proposed that endometriosis can be differentiated from embryological remnants. The mechanism that endometriotic lesions originate “in-situ” by metaplasia or from Müllerian remnants further explains the endometriosis in adolescents shortly after menarche and in fetuses with the absence of menstruation [11,12,13].

### 2.2. Circular Dissemination and Stem Cell Theory

For uncommon extraperitoneal locations, Sampson suggests that the shedding cells enter the uterine vasculature or lymphatic system, disseminating to distant organs, including the lung, liver, spleen, and brain [8,14,15]. An alternative theory has become clear that stem cell contributes to the pathogenesis of endometriosis [16]. Bone marrow-derived stem cells (BMDSCs) travel to the uterine cavity and regenerate eutopic endometrium. During menstruation, women with endometriosis have more pluripotential cells and shed more progenitor cells than healthy women, further expanding the hypothesis of retrograde menstruation [17,18]. These BMDSCs can directly differentiate into endometrial cells at ectopic locations in the peritoneal cavity and distant sites [19,20].

### 2.3. Invagination Theory in Adenomyosis

Endometriosis of uterus is characterized by a junctional zone disease, resulting from altered endometrial basalis cells invading the uterine myometrium through disruption of the endometrial-myometrial interface (EMI), subsequently establishing ectopic endometriotic lesions [21,22]. Higher risks of adenomyosis are found in women with microtrauma of EMI, such as repeated endometrial curettage, cesarean delivery, and previous uterine surgery [23]. Furthermore, cycles of sustained uterine hyperperistalsis and repeated myocyte overstretching augment auto-traumatization, potentially promoting the invagination of endometrial basalis in the tissue healing process [24,25,26] (Figure 1).

### 2.4. Epigenomic and Genomic Alterations

Several studies have reported the relationship between heritability and endometriosis. The results of family aggregation studies suggest that the probability of developing endometriosis is 8% from a diseased mother and 6% from an affected sister [27,28]. The risk of developing the disease is less than 1% in both situations in the control population without a family history. The increased disease prevalence has been found in first-degree relatives of women with endometriosis, although the specific genetic origin of the association remains unknown. Genome-wide association studies (GWAS) have investigated samples from thousands of women in the United States, Australia, Europe, and Japan [29,30,31]. These new tools have identified genetic variations (Single Nucleotide Polymorphisms, SNPs) that are thought to be missense mutations in patients with endometriosis, among which the following genes involved in the development of endometriosis through cell proliferation, migration, and adhesion: VETZ, WNT-4, GREB1, CDKN2B-AS1 and ID4. VETZ participates in cell adhesion, migration and transmembrane cell junction, which is associated with epithelial mesenchymal transition; WNT-4 plays decisive roles during the development of female reproductive system; GREB1 is responsible for estrogen regulation; CDKN2B-AS1, a tumor suppressor gene and ID4, an ovarian oncogene have been implicated in molecular pathogenesis of endometriosis. The specific loci strongly correlate SNPs with advanced disease development [28,32,33,34]. Unsurprising, the common SNPs associated with endometriosis have been detected overlap compared to other gynecological disorders such as infertility, fibroids, and cancer [35,36,37]. The similar loci between SNP datasets implicate the shared pathogenesis of other gynecologic diseases that the genes significantly associated with endometriosis risk are close to genes involved in sex steroid hormone pathways [38], MAP kinase signaling cascade [39], interleukin 1A (a cytokine implicated in inflammatory responses) [31], WNT signaling [36], and steroid metabolism [38]. With regard to genetic markers obtained from GWAS to identify high-risk populations for developing endometriosis, each functional group involved in pathogenesis has at least one or more genes that link with endometriosis. Nevertheless, until now, no genetic tests could be considered reliable for clinical diagnosis; translating findings into validated tests should only be performed within a research setting [40].

Studies on the impact of epigenetic modifications expanded rapidly to date. Without altering the DNA sequence, these regulatory mechanisms modify gene expression through DNA methylation, histone modification, and noncoding RNA regulation [34]. The first discovered evidence showed that the promoter of the HOXA10 gene is hypermethylated in women with endometriosis compared to healthy individuals. Methylation of the Homeobox A (HOXA10) leads to gene silencing, which reduces the expression of E-cadherin, favoring the process of cell invasion from the weak intercellular junctions [41,42]. Additionally, progesterone resistance is generated by hypermethylation of the progesterone receptor-β (PR-β), contributing to compromised stromal epithelial crosstalk [43,44,45]. Some hypomethylated genes, including estrogen receptor-β (ER-β), steroidogenic factor-1 (SF-1), and aromatase, are overexpressed, increasing estrogen and its receptor, respectively [46]. 

Histone modification, characterized by changes in chromatin structure, provides or blocks access of effectors modulators and of transcription factors to their binding sequences in gene promoters, leading to gene deregulation and disease. Generally, histone methylation causes gene silencing while histone acetylation contributes to activation of gene expression. Post-translational histone modification within promoter regions of selected genes has been reported in endometriosis [47,48]. Hypoacetylation of H3/H4 within promotor regions of target genes known to be downregulated in endometriosis, such as HOXA10. Hyperacetylation of H3/H4 within promotor regions of SF-1, a transcription factor involved in estrogen biosynthesis, results in overexpression of aromatase and local estradiol. These data contribute to a better understanding of gene expression regulated by histone modification.

Noncoding RNAs (ncRNAs), which are not translated into a protein, serve to regulate chromosome structure, interact with messenger RNA, and usually inhibit gene expression at a post-transcriptional level [49]. MicroRNAs (miRNAs) are small single-stranded ncRNAs. The endometrium’s miRNA signatures change with the different phases of menstrual cycles [50,51]. Hundreds of dysregulated miRNAs have been found in paired eutopic and ectopic endometriotic lesions of women with and without endometriosis [52,53]. Extracellular miRNAs are found in circulation [54] and can potentially mediate intercellular communication between eutopic endometrium and ectopic endometriotic lesions [55]. Up- or down-regulation of miRNA plays a crucial role in the endometriotic implant establishment [56,57,58] and provides potential therapeutic targets in the future.

### 2.5. Estrogen and Progesterone Modulation

In both normal endometrium and ectopic endometriotic lesions, steroid hormones and their receptors regulate cell proliferation, angiogenesis, neurogenesis, and inflammatory pathways. Properly balanced concentrations of estrogen and progesterone regulate functional eutopic endometrium during menstrual cycles. Endometriosis is also defined as a ‘‘steroid-dependent’’ disorder, which depends on its cell-specific patterns of steroid receptor expression and menstrual cycle phase-dependent hormone metabolism [9,44]. Estradiol activates cyclooxygenase-2 (COX-2) within uterine endothelial cells, increasing prostaglandin E2 production in a feed-forward mechanism [59]. An increase in estradiol production by aromatase, the loss function of 17-hydroxysteroid dehydrogenase type 2 (17β-HSD2), and the overexpression of ER-β promote cell growth and perpetuated inflammation in ectopic endometriotic lesions [60,61,62]. The accumulation of estradiol activates mitogenic activity by stimulating a series of genes related to cell proliferation (GREB1, MYC, and CCND1), inhibiting the apoptosis induced by apoptosis signal-regulating kinase-1 (ASK-1) and tumor necrosis factor-α (TNF-α), leading to the development of endometriotic implants [63]. 

In contrast, downregulation of PR-β results in progesterone resistance, which causes a nonreceptive eutopic endometrium and a pro-inflammatory microenvironment limiting the effects of progestin therapy and further driving the systemic impact of endometriosis [34,64]. With regard to strong progesterone resistance in endometriotic lesions, the effect of hormone therapy on superficial endometriotic lesions seems to be a consequence of the decreased estrogen concentrations rather than a direct progestin effect [65]. The optimal solution would be partially reducing estrogen levels just enough to suppress survival and vascularization of endometriotic implants, while at the same time maintaining adequate concentrations to alleviate vasomotor menopausal symptoms and bone mineral density loss.

### 2.6. Inflammation, Angiogenesis, and Tissue Remodeling

Endometriotic implants are complex multicellular structures that ectopic endometrial cells migrate, adhere, and evade through a serial process of tissue remodeling, followed by the influx of pro-inflammatory cytokines and the growth of new blood vessels (angiogenesis) [66] (Figure 2). Peritoneal fluid in affected patients is also found to contain increased pro-inflammatory cytokines [67,68,69]. The aberrantly increased concentrations of interleukins (IL-1β, IL-6, IL-8, IL-33), tumor necrosis factor-alpha (TNF-α), insulin-like growth facto-1 (IGF-1), monocyte chemoattractant protein/C-C motif chemokine ligand (MCP-1 CCL2, CCL5), and vascular endothelial growth factor (VEGF) activate the inflammatory response by upregulating nuclear factor kappa-light-chain-enhancer of activated B cells (NF-κB) in affected women [70,71,72,73,74,75,76,77]. Circulating cytokines and immune cells further create a widespread inflammatory environment which drives the systemic effect of endometriosis on immunologic, cardiovascular, neurological, and metabolic function [64,71,76,78,79]. VEGF/tyrosine kinase signaling pathway has been upregulated and involved in numerous mechanisms of vascularization, including de novo growth (angiogenesis), vasculogenesis, and the formation of interconnected networks. Furthermore, the link between the growth of new blood vessels and nerve fibers contributes to the “neuro-angiogenesis”, ectopic endometriotic lesions, and pain pathways [80,81].

Stromal fibroblasts, with the phenotype of clonogenic and multilineage potential, contribute to multicellular lesions at extra-uterine locations [45,82]. In women with an aberrant response to estrogen, the endometriotic lesions formed from tissue fragments regulates by transient hypoxia [83], the release of iron, and the activation of platelets [84,85].

### 2.7. Immune Dysregulations

Aberrant production of pro-inflammatory cytokines recruits a large pool of immune cell populations which alters the peritoneal environment in women with endometriosis [1,4,86]. The abundance of innate immune cells and different populations of adaptive immune cells has been detected in the peritoneal fluid of affected women or the endometriotic lesions from patients implicating a compromised immune system in endometriosis [1,87,88]. The neutrophil chemotactic factors such as IL-8, granulocyte colony stimulating factor (G-CSF), and chemokine ligands 1, 2, and 3 (CXCL-1, CXCL-2, and CXCL-3) gather the immune cells in a loop of positive feedback. In peritoneal fluid and eutopic endometrium of women with endometriosis, a significant increase of macrophages has found a decrease in the phagocytic activity [89,90], which promotes angiogenesis [91], lesion innervation [68,91] and pain symptoms [92]. Activated macrophages are divided into two phenotypes, among which M1 macrophages are dominant in pro-inflammatory responses and M2 macrophages are mainly involved in anti-inflammatory responses. Although the studies remain controversial, the affected women have both physiological endometria with M1 predominates and ectopic portions with M2 polarization, allowing angiogenesis, tissue remodeling, and thus the development of the disease [32,86,90,93,94]. Survival of ectopic lesions is also provided by the activation of T and B cells and the decreased cytotoxicity of natural killer (NK) cells [95,96]. The cytokines, including the platelet-derived transforming growth factor β (TGF-β), IL-6, and IL-15, inhibit the cytotoxicity of NK cells, thus contributing to the implantation, proliferation, and immune escape of ectopic endometrial cells [95,96,97].

## 3. Clinical Features of Endometriosis and Its Lifelong Impacts

Endometriosis causes heterogeneous presentation, varying from superficial peritoneal lesions, ovarian(endometrioma) and uterine(adenomyosis) tumors, and deep infiltrative endometriosis (DIE), which is often accompanied by scarring and adhesions [1,4]. These lesions are associated with gynecological disorders and the variability of pain symptoms. Endometriotic lesions can also spread and grow in extra-pelvic sites, including visceral organs in the upper abdomen, chest (thoracic endometriosis), brain, and nerve systems [98] (Figure 3). Although none of the biomarkers displayed enough accuracy, understanding the diffuse clinical presentations and diverse disease patterns of endometriosis helps early identification and intervention.

### 3.1. Menstrual Disorders in Adenomyosis

Gynecologic disorders such as dysmenorrhea, menorrhagia, and abnormal uterine bleeding are the main symptoms in women with adenomyosis. In adenomyosis-affected myometrium, higher expression of oxytocin receptors and altered membrane depolarization of uterine smooth muscle cells contribute to abnormal uterine contractility [99,100,101]. The increased expression of TNF-α promotes the production of IL-1β [102], activation of the NF-κB pathway [103], and engagement of IL-18/IL-18R complex [104,105], resulting in pain symptoms by mediating prostaglandins synthesis. The severity of dysmenorrhea also correlates with immunoreactivity, neuropathologic factor, and microvascular function [106]. Abnormal uterine contractility and high microvessel density further cause heavy menstrual bleeding.

### 3.2. Endometriosis-Associated Symptoms

The common symptoms of pelvic endometriosis are chronic pelvic pain (cyclical and non-cyclical) and other pain conditions, including painful sexual intercourse (dyspareunia), painful urination (dysuria), and painful defecation (dyschezia). The severity of pelvic endometriosis can be determined after surgical intervention using the revised scoring system of the American Society for Reproductive Medicine (ASRM) [107] (Appendix A). The stage of pelvic endometriosis is not always correlated with patient-reported pain symptoms [108], and a lack of awareness of deep infiltrative endometriosis and extra-pelvic endometriosis may delay diagnosis. Some women experience recurrent, particularly perimenstrual, changes in bowel habits (diarrhea or constipation), irritable bowel syndromes, and bloody stool implicating the evidence of deep infiltrative endometriosis of the lower gastrointestinal tract. Other women suffering from cyclic dysuria and hematuria have been treated as refractory urinary tract infections or bladder pain syndrome, which can be caused by endometriosis [109]. Endometriosis of the diaphragm and pleura has been associated with chest and shoulder pain [110]. Neurological changes and chronic inflammation in endometriosis enhance pain perception, anxiety, fatigue, and depression [111,112]. 

### 3.3. Endometriosis-Associated Infertility

Endometriosis should be considered a cause of infertility in women with pain symptoms. Avoiding sexual intercourse due to severe dyspareunia and chronic pelvic pain limits the feasibility of natural conception [113,114]. Pelvic adhesion can also cause anatomical distortion, interrupting the conception process, including oocyte release from ovaries, ovum pickup, and transport through fallopian tubes [115]. Secondly, oxidative stress in an endometrioma causes damage to the adjacent healthy ovarian cortex, reducing ovarian reserve [116,117]. Alterations of the intrafollicular microenvironment and aberrant steroidogenesis impair folliculogenesis and oocyte competence [118,119]. Dysregulation of immune and inflammatory profiles plays an essential role in recurrent implantation failure and early pregnancy loss [49,120,121]. Adenomyosis, characterized by the defective junctional zone and perturbed uterine peristalsis, is strongly associated with primary infertility and adverse in vitro fertilization (IVF) outcomes [122,123].

### 3.4. Endometriosis-Associated Obstetric Complications

Emerging research has demonstrated the relationship between endometriosis and obstetric complications, including miscarriage, preterm birth, preterm premature rupture of membranes, antepartum hemorrhage, placental abruption, placenta previa, preeclampsia, gestational hypertensive and metabolic disorders (diabetes or cholestasis), and adverse neonatal outcomes (small for gestational age, low birth weight, admission to neonatal intensive care and neonatal death) [124,125]. Endometriosis and obstetric diseases share some molecular features and pathophysiologic mechanisms of the defective junctional zone, perturbed uterine peristalsis, and aberrant inflammation. Several differentially expressed genes involved in endometriosis are common in adverse pregnancy outcomes such as preeclampsia, small for gestational age, or preterm birth. Alterations of imprinted gene clusters (CDKN1C, DLX5, GATA3) in the link between endometriosis and abnormal decidualization are considered critical regulators of embryogenesis and placentation [126,127,128,129,130,131,132]. Adverse maternal environments can lead to placental genetic and epigenetic aberrant, which alters the placenta’s ability to modulate fetal exposure and response to maternal cortisol, causing infant neurobehavioral deficits [133]. In addition to suboptimal placentation, overexpression of COX-2 and prostaglandins secretion in chronic inflammation can lead to early cervical ripening and uterine hypercontractility in women with endometriosis, thus causing adverse fetal outcomes [134,135,136]. 

### 3.5. Malignancy Potential

A recent meta-analysis has reported that endometriosis is associated with an increased risk of certain cancers, such as ovarian, breast, and thyroid cancers [137]. Endometriosis induces the microenvironment with an aberrant immune response and altered hormonal milieu [60], favoring neoplastic transformation. Accumulations of oxidative stress and chronic inflammatory response contribute to the development and progression of endometriosis-associated malignancies. Somatic mutations in the genes of women with endometriosis have been recognized as a precursor of malignant transformation [138]. Several genetic studies have discovered that mutations or alterations in genes (PTEN, TP53, KRAS, and ARID1A) of endometriosis are directly related to neoplasms [139,140,141,142,143]. Atypical endometriosis, as a histologically borderline tumor corresponding to the features of hyperchromatic nuclei, cellular crowding, and the high nucleus-to-cytoplasm ratio [144,145], has a greater risk of malignant change to clear cell and endometrioid ovarian cancers [146]. Recent evidence of a link between endometriosis and malignant potential has raised concerns in the long-term management of patients with endometriosis through the lifetime from puberty to post-menopause [147].

### 3.6. Long-Term Systemic Diseases

Endometriosis, rather than a localized pelvic disease, has a detrimental effect on cardiovascular, neurological, metabolic, and immune function, stemming from circulating pro-inflammatory cytokine and shifts in immune cell populations [148]. Increasing research has reported that differential gene expression in endometriosis alters metabolism in the liver and adipose tissue leading to systemic inflammation [149,150,151] but also affects gene expression in the brain causing pain sensitization, anxiety, and depression [111,152,153]. These data suggest that a variety of metabolic phenotypes in endometriosis results in a life course of systemic effects. Early recognition and management of all aspects of the disease can relieve problematic symptoms and avoid long-term sequelae.

## 4. Pharmacologic Therapies in Current Clinical Practice

Clinical management of endometriosis-associated symptoms depends on the disease’s severity, extent, and location. The choice of treatments includes medication, surgery, or a combination of both. Pharmacological therapy for endometriosis aims to relieve symptoms, maintain long-term control, or prevent recurrence after surgical removal of lesions. However, therapeutic windows in the life course of endometriosis are challenging due to unwanted side effects and the desire to conceive.

### 4.1. Hormonal Manipulation

Hormonal targets on presumed altered steroidogenesis in endometriosis act by suppressing fluctuations in gonadotropic and ovarian hormones, establishing either a hypo-estrogenic or hyper-progestogenic milieu, resulting in the inhibition of ovulation and the reduction of menstrual bleeding [9,154] (Figure 4).

#### 4.1.1. Progestin-Based Therapies

Combined oral contraceptives (COCs), containing estrogen and progestin, have been extensively prescribed in clinical practice due to their efficacy in managing dysmenorrhea. It is particularly beneficial in continuous rather than cyclic administration [1,40,156,157]. Progestin-dominant COCs can establish a hyper-progestogenic status, inducing decidualization and subsequent apoptosis of ectopic endometriotic implants [158]. The estrogen component induces central inhibition of gonadotropin secretion, inhibiting ovulation and reducing ovarian estrogen production, creating a hypo-estrogenic milieu [139]. However, the main concern about the off-labeled use of COCs in endometriosis is that the estrogen content of COCs potentially contributes to the progression of endometriosis [159,160].

Progestins [155], acting as natural progesterone, can induce anovulation and endometrial pseudo-decidualization, resulting in the atrophy of endometriotic implants by decreasing inflammation and angiogenesis [161,162]. Progestin-only pills, including dienogest, norethisterone, and medroxyprogesterone, are currently the first-line treatment for symptomatic endometriosis and aim to prevent recurrence after surgery [163,164,165]. Additionally, progestins can be administered by other routes such as intramuscular, subcutaneously (etonogestrel implant), or intrauterine (levonorgestrel-releasing intrauterine device, LNG-IUD) [166]. Dienogest (2 mg daily), a 19-nortestosterone derivative, can increase PRβ expression in endometriotic lesions, potentially overcoming progesterone resistance [167]. Several randomized controlled trials have proved its efficacy for endometriosis-associated pain regarding different phenotypes [165,168,169,170]. Dienogest may decrease the size of ovarian endometrioma without decreasing ovarian reserve [171,172,173] and reduce pain symptoms related to deep infiltrating endometriosis (dysmenorrhea, dyspareunia, dyschezia, and chronic pelvic pain), thus improving patients’ quality of life. Norethisterone acetate (NETA, 2.5–15 mg daily), a 19-nortestosterone derivative, was confirmed as effective as dienogest in reducing the size of ovarian endometrioma and endometriosis-related symptoms, whereas dienogest was superior in symptoms relief and tolerability [172,174,175]. 

Medroxyprogesterone acetate (MPA, 10–60 mg daily), a 17OH-progesterone derivative available as oral or depot formulation (administered every three months subcutaneously or intramuscularly), is as effective as Gonadotrophin releasing hormone analogs (GnRH agonist) and limited to treat refractory endometriosis due to long-term hypo-estrogenic status, consequently leading to bone loss [176,177]. LNG-IUD, a potent 19-nortesterone derivative released directly into the uterine cavity, improves menstrual disorders and pelvic pain symptoms related to endometriosis [178,179,180]. 

Commonly reported side effects of progestin-based therapies are abnormal uterine bleeding, which progressively improves with the continuation of treatment, headache, mood changes, and particularly for long-term use of depot MPA, as well as loss of bone marrow density [181].

#### 4.1.2. GnRH Agonists

GnRH agonist, substituting a D-amino acid for the native L-amino acid at position six of the native GnRH peptides, initially stimulates gonadotrophins secretion (the flare effect) and subsequently reduces estrogen production by downregulation and desensitization of the pituitary GnRH receptors [182]. The induced hypo-estrogenic status can cause the regression of endometriotic lesions, but the prolonged receptor occupancy leads to vasomotor symptoms, vaginal dryness, reduced libido, sleep disturbance, mood disorder, and bone loss [183]. Add-back therapy (the addition of progestin alone or COCs) has been advocated to extend pain relief up to 10 years of treatment by minimizing the adverse effects of estrogen deprivation [184,185,186]. However, GnRH agonist remains a second-line treatment for endometriosis because of the high cost and the limitation of long-term maintenance [148].

#### 4.1.3. GnRH Antagonists

GnRH antagonists, similar in structure to natural GnRH, suppress pituitary function through direct competition with GnRH receptors and thus induce a rapid drop of estrogen without provoking the flare effect [187]. Upcoming clinical trials have explored the efficacy and safety of oral nonpeptidic GnRH antagonists, including Elagolix, Relugolix, and Linzagolix, for treating endometriosis-associated pain [188]. Elagolix (150 mg daily to 200 mg twice daily), a uracil derivative, the first approved compound, is effective in reducing moderate and severe endometriosis-associated pain for six months of treatment in two phase III trials (Elaris Endometriosis I and II) [189] and improving quality of life for six additional months in phase III extension studies (Elaris EM-III and EM-IV) [190,191,192]. Relugolix (10–20–40 mg daily), a thieno [2,3-d]pyrimidine-2,4-one derivative, alleviates endometriosis-associated pain in a dose-response manner for the 12-week treatment in a Phase II trial [193] and with similar results for 24-week therapy in two other replicate phase III trials (SPIRIT-1 and 2) [194,195]. Linzagolix (75–100–200 mg daily), the newest compound of GnRH antagonists, has been investigated for 24-week treatment in a phase IIb trial [196] and in the extension study up to 52 weeks [197], which showed promising therapeutic effects on pain symptoms in women with endometriosis. Like GnRH agonists, the side effects of GnRH antagonists, such as hot flushes and bone loss, are related to the hypo-estrogenic status and proportional to the doses and duration of treatment [189,191,195,196].

#### 4.1.4. Other Potential Hormonal Drugs

Other drugs currently under investigation include aromatase inhibitors (AI, e.g., Anastrozole, Letrozole), selective progesterone receptor modulators (SPRM, e.g., Mifepristone, Anoprisnil), or selective estrogen receptor modulators (SERM, e.g., Bezedoxifene, Raloxifene). Considering the high rate of adverse effects, guidelines from the European Society of Human Reproduction and Embryology (ESHRE) recommend that these drugs be used in a scientific setting or combined with other medicines for women refractories to other traditional hormonal treatments [40,156].

Hormonal suppressive therapies modify the endocrine environment in both eutopic endometrium and ectopic lesions, reduce menstrual bleeding that will decrease retrograde flow, and blunt the triggering of inflammatory pathways implicated in menstrual pain [198]. Although the endometriotic lesions histologically resemble the endometrial lining of the uterus, the biochemical differences between eutopic endometrium and endometriotic tissue can be interpreted as the consequence of cell-specific patterns of steroid receptors expression, the immunological microenvironment, questioning the similarity of endometrium to endometriotic lesions [65]. Most subtle superficial endometriosis lesions must have a strong progesterone resistance, considering high progesterone concentrations in peritoneal fluid [199]. The mechanisms of resistance include being implanted basal endometrium, an effect of peritoneal fluid, and epigenetic changes in endometriotic lesions [65]. However, epigenetic therapies are far from ready for clinical application in patients with endometriosis [200]. Peritoneal fluid might be important for understanding the initiation and growth of endometriosis lesions and the lack of synchronicity with the endometrium. Higher estrogen and progesterone concentrations are found in the peritoneal fluid compared with plasma both in women with and without endometriosis [201]. After ovarian suppression, the decreased estrogen concentrations in peritoneal fluid are probably comparable to plasma concentrations which are very low, thus reducing endometriotic lesions [65].

### 4.2. Analgesics and Neuromodulators

Hormonal therapy for the first-line treatment is typically accompanied by direct analgesia using nonsteroidal anti-inflammatory drugs (NSAIDs), paracetamol (acetaminophen), or various opioids [202]. NSAIDs, widely used in treating chronic inflammation, are effective in relieving pain symptoms. They inhibit the cyclooxygenase enzymes, thus reducing prostaglandin production and inflammation. Common side effects such as gastric ulcers, cardiovascular events, and acute kidney injury should be concerned for long-term use. Increased pain perception in the endometriotic lesions and the nervous system has been demonstrated in women with chronic pain symptoms [203]. The mechanisms that overexpressed nociceptive channels, including P2X3, small diameter sensory neurons, and increased neuropeptides, including calcitonin gene-related peptide, substance P, and neurokinin, play an essential role in generating hyperalgesic responses [204,205,206]. A new investigational drug inhibiting purinergic P2X3 receptor (BAY1817080) has been recruited in clinical trials (Table 1).

### 4.3. Targeting on Inflammation, Angiogenesis, and Immunomodulators

IL-1 cytokine family (including IL-33 and IL-1β) and their receptors show potential targets for inhibition of inflammation by downregulating the MyD88 pathway, reducing endometriotic lesions [69,207]. MT2990, a monoclonal antibody directly against IL-33, appears to have promising results in treating moderate to severe endometriosis-associated pain (NCT03840993). Another exploratory trial (NCT03991520) also shows the therapeutic potential of Anakinra, an IL-1 receptor antagonist in endometriosis. Anti-angiogenic agents (e.g., DLBS1442, Quinagolide, and Cabergoline) involved in inflammatory or pain pathways also have promising therapeutic effects on endometriosis. However, a previous study using anti-TNF-α therapy (Infliximab) failed to show a welcome impact on women with deep endometriosis [208].

### 4.4. Other Complementary Therapies

Dietary supplements and natural products, such as Omega-3 polyunsaturated fatty acids, resveratrol (derived from grapes/berries), curcumin (derived from the roots of Curcuma longa), and green tea (rich in catechins/flavonoids) are considered complementary therapies for endometriosis due to their antioxidative, antimitotic, anti-inflammatory, and anti-angiogenic properties [209,210,211,212]. These bioactive compounds reduce the expression of IL-6, IL-8, TNF-α, and COX-2, presenting a reduction of VEGF expression and matrix metalloproteinase-9 activity, thus inhibiting the development of endometriosis [213,214].

## 5. Surgical Considerations

While medical therapies are not always effective in specific subtypes of endometriosis, surgical excision of all visible lesions is considered the alternative treatment for refractory pain symptoms and intolerable adverse effects of medical therapy [202,215,216]. Surgical approach helps define the severity of endometriosis, restore the pelvic anatomy, and obtain the tissue specimen to rule out suspected malignancy. Surgical reduction of ovarian endometrioma has raised attention to the damage to ovarian reserve and addressed debated issues on managing ovarian endometrioma. DIE can cause irreversible fibrosis and pelvic adhesion, obstructing bowels, ureters, and fallopian tubes [108]. Completely surgical destruction of DIE has higher complication rates, particularly when colorectal and urological resection and/or anastomosis is concomitantly required [217,218,219,220]. Conservative surgery, known as fertility-sparing/preservation surgery (resection of endometriotic lesions without removal of the ovaries and the uterus), is preferred in women with a desire for pregnancy, retaining natural fertility, and augmenting assisted conception [156]. Long-term medical maintenance following conservative surgery reduces endometriosis recurrence, thus avoiding repetitive surgery [221,222,223,224]. In addition, there is increasing awareness of persistent endometriosis-associated pain following surgery [225], exacerbating pain symptoms after repeated surgery, and chronic post-surgical pain (CPSP), which occurs in the postoperative 3 to 6 months [226]. These sequelae imply concomitant adenomyosis, chronic central pain sensitization, or other non-endometriotic-related problems [227]. Accordingly, surgery should be considered only when the benefits overwhelm the drawbacks.

## 6. Management of Endometriosis-Related Infertility

Women with endometriosis often experience infertility either because of endometriosis itself or due to hormonal suppression [228]. Most of the medical therapies currently used for endometriosis prevent or compromise conception; therefore, reproductive counseling and fertility survey should be established prior to surgery and at diagnosis [40]. The therapeutic decision-making strategies for the complexity of endometriosis are presented in Figure 5.

### 6.1. Fertility Treatments and Assisted Reproductive Technology (ART)

For women who immediately attempt conception, expectant management is an option but is not recommended due to the low fecundity rate [229,230]. Ovulation induction combined with intrauterine insemination increases the pregnancy rate in women with mild endometriosis [231,232]. In vitro fertilization (IVF) remains the most effective treatment of endometriosis-associated infertility [233,234]. Hormonal suppression by GnRH agonist for 3–6 months or continuous COCs use for 6–8 weeks before IVF has been reported to increase the pregnancy rate, probably due to improving the endometriosis-associated hormonal disturbance [235,236,237,238,239]. Based on evidence from RCTs, a meta-analysis of studies comparing different GnRH agonist protocols (short, long, ultra-long) reported that, a GnRH agonist ultra-long protocol could improve clinical pregnancy rates, especially in patients with stages III/IV endometriosis. However, considering both RCTs and observational studies, the different downregulation protocols showed no significant difference in improving clinical outcomes in patients with endometriosis [240]. In a retrospective study including 151 patients with endometriosis and a previous failed IVF cycle, DNG pre-treatment for 3 months prior to IVF versus no pre-treatment significantly increased cumulative implantation, clinical pregnancy, and live birth rates [241]. This controversial evidence is insufficient to recommend the extended administration of GnRH agonist prior to ART treatment [40]. Current studies showed promising results in a segmented ART protocol that initiates an IVF with all embryos frozen and then administers GnRH agonist or progestin for 3–6 months, followed by deferred thawed embryo transfer [242,243] (Figure 6). Controlled ovarian stimulation may also accelerate endometrioma growth, causing mass effects that diminish oocyte maturation and ovum pickup. However, there are conflicting results regarding the benefits of endometrioma excision before ART [244,245,246].

### 6.2. Fertility Preservation in Patients with Endometriosis

For women who do not immediately desire pregnancy, endometriosis and ovarian surgery have an increased risk of diminished ovarian reserve [229,230]. Available procedures for fertility preservation include embryo or oocyte freezing and ovarian tissue cryopreservation, which is no longer considered experimental technology [247,248,249]. Generally providing fertility preservation could expose affected women to unnecessary iatrogenic risks, and the effectiveness remains controversial [250,251,252,253]. Women with bilateral endometrioma, repetitive ovarian surgery, or known diminished ovarian reserve are optimal candidates for fertility preservation after reproduction counseling [250,254].

## 7. Conclusions

Endometriosis was previously considered a gynecologic disease associated with menstrual disorders, pelvic pain symptoms, and infertility. Following the understanding of endometriosis pathogenesis, the systemic and long-term impacts of endometriosis have raised concern. Management of patients in whom endometriosis is found incidentally (without pain or infertility), adolescents and menopausal women with endometriosis depend on better understanding of the disease diversity and associated risks, such as malignant transformation, cardiovascular and metabolic disease. Hormonal regulation, angio-neurogenesis and inflammatory pathway play a crucial role in the complex pathogenesis of endometriosis. The mechanisms involved in endometriosis are not fully discovered, and managing long-term endometriosis-associated complications is still challenging. The current strategy of endometriosis aims to relieve associated symptoms, avoid repetitive surgery, preserve fertility, and potentially reduce the lifelong systemic impacts. Hormonal modification remains the mainstay of endometriosis treatment at initial diagnosis and for long-term maintenance. However, symptomatic women who desire to become pregnant have limited therapeutic options. In light of the limited efficacy and intolerable side effects of many commonly used drugs, future treatment modalities focusing on new molecular targets are most urgently required to enhance lifespan management of the full scope of endometriosis.

## Figures and Tables

**Figure 1 ijms-24-07503-f001:**
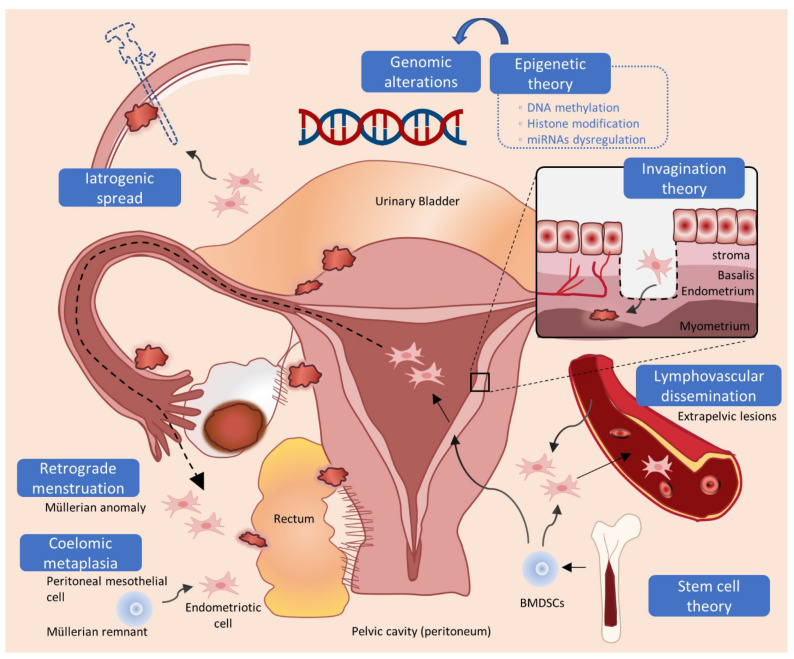
Theories of endometriosis pathogenesis. The potential origins of endometriotic lesions include traveling of endometrial tissue through retrograde menstruation and in situ by coelomic metaplasia of the peritoneal lining or from Müllerian remnants. Lymphovascular dissemination contributes to extra-pelvic lesions. Bone marrow-derived stem cells (BMDSCs) can directly differentiate into endometrial cells at ectopic locations in the peritoneal cavity and distant sites, further expanding the hypothesis of retrograde menstruation and lymphovascular dissemination. Invagination theory is characterized by altered endometrial basalis cells invading the uterine myometrium through disruption of the endometrial-myometrial interface (EMI), resulting in endometriosis of uterus. Epigenomic and genomic alterations further explain the aberrant gene expression in endometriotic lesions.

**Figure 2 ijms-24-07503-f002:**
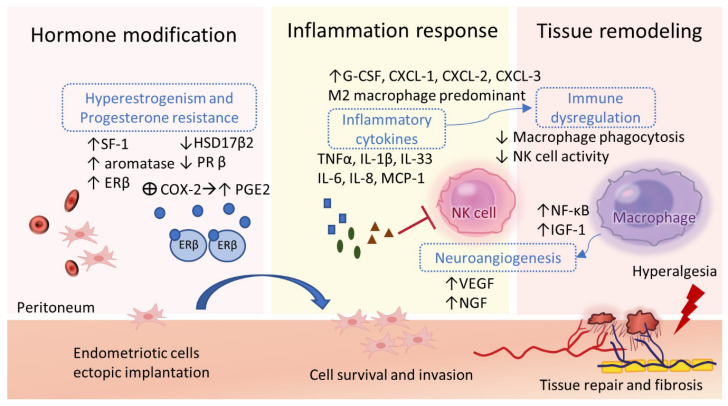
Pathophysiological processes of endometriosis. Endometriotic lesions are established through interacting molecular mechanisms in a micro-environment of hyperestrogenism and progesterone resistance that promote cell survival and invasion, systemic and localized steroidogenesis, inflammatory response, immune dysregulation, and neuro-angiogenesis. The upward arrows represent overexpression; the downward arrows represent down-regulation.

**Figure 3 ijms-24-07503-f003:**
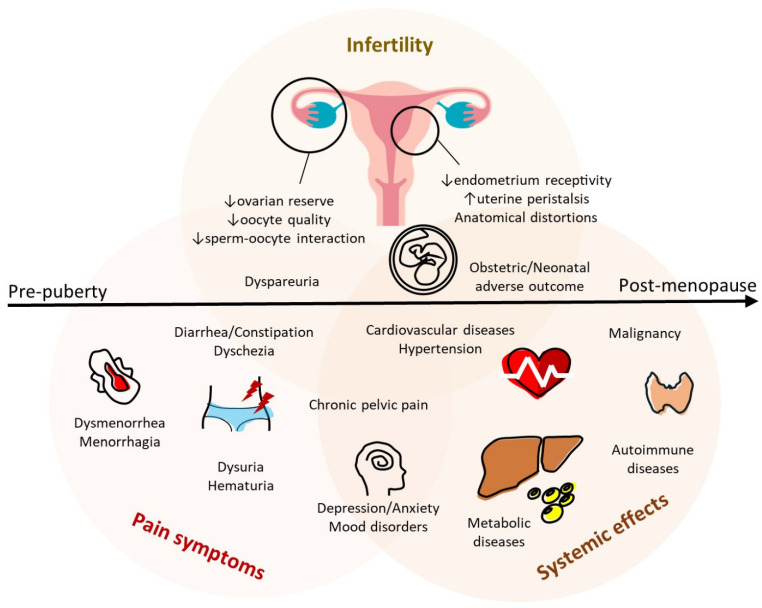
Endometriosis across the life course. Summary of the symptoms associated with endometriosis, including pain, infertility, and systemic effects. They have a wide range of spectrum and can overlap with these other conditions. The upward arrows represent an increase, and the downward arrows represent a decrease.

**Figure 4 ijms-24-07503-f004:**
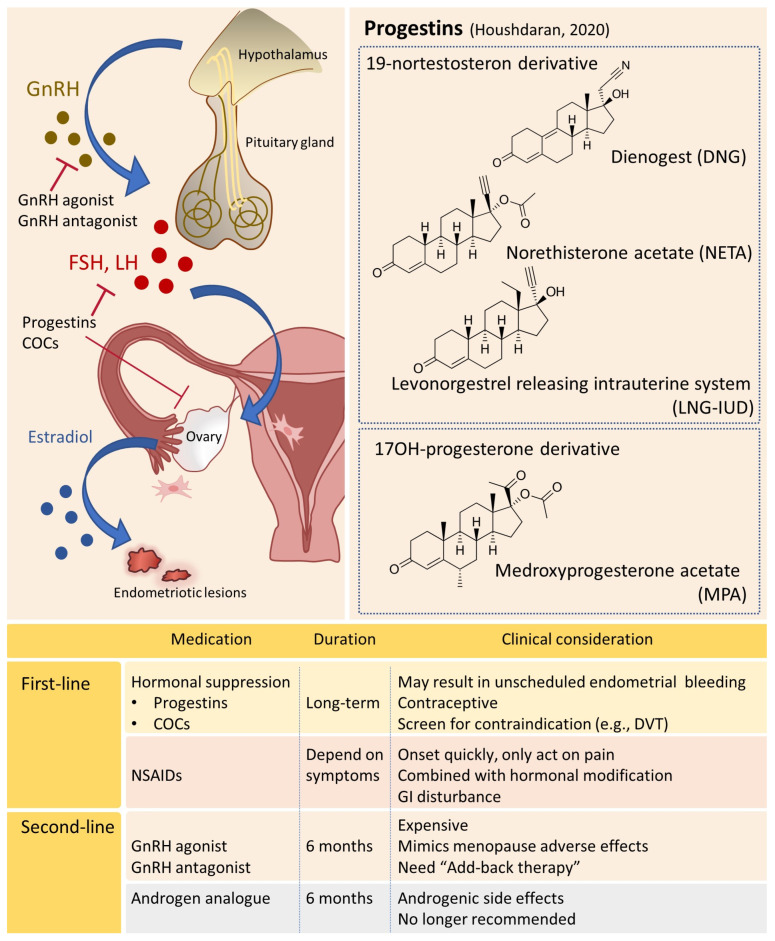
Medical treatments for endometriosis [155].

**Figure 5 ijms-24-07503-f005:**
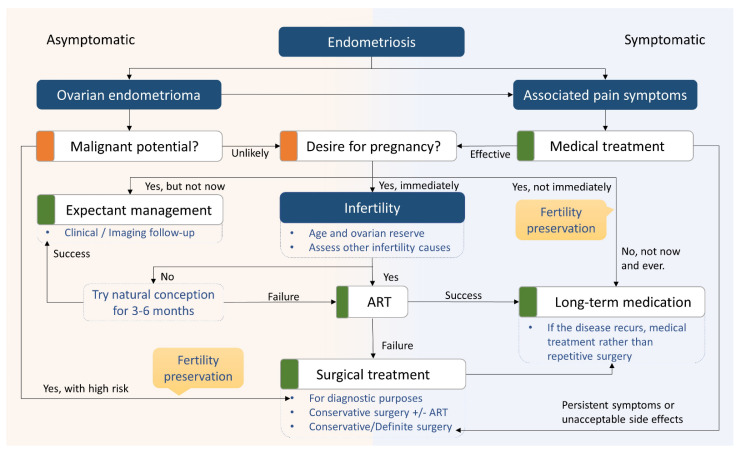
Treatment algorithm of endometriosis.

**Figure 6 ijms-24-07503-f006:**
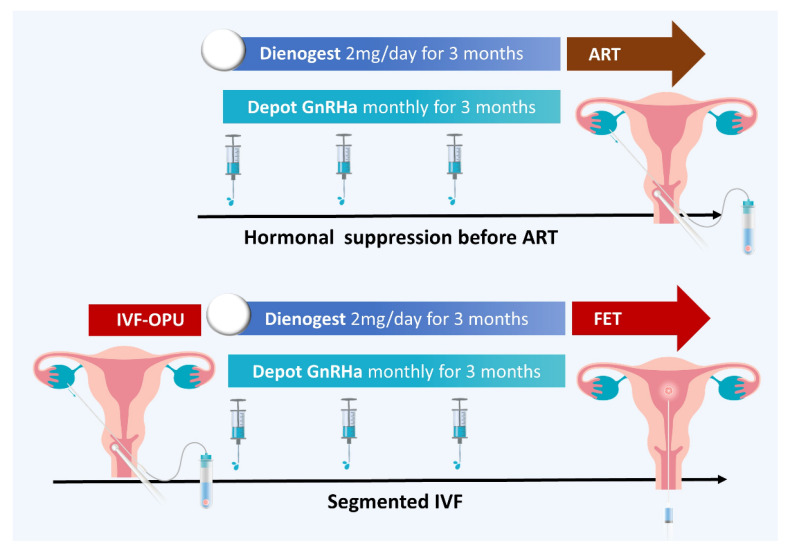
Assisted reproductive technology for endometriosis.

**Table 1 ijms-24-07503-t001:** Investigational drugs for endometriosis.

Mechanism	Targets	Investigational Drugs	Clinical Trials	Status
Anti-angiogenic drugs	Selective dopamine D2-receptor agonist	Quinagolide (vaginal rings) Cabergoline (oral)	NCT03749109 NCT03692403 NCT03928288	Phase II completed Phase II completed Phase II recruiting
Immunomodulators	Suppression NF-ᴋB and COX-2	DLBS1442 (oral)	NCT01942122	Phase II completed
Antioxidants	Downregulates inflammatory cytokines (e.g., IL-6, IL-1β, MCP-1)	Resveratrol (oral)	NCT02475564	Phase IV completed
Anti-neurogenic and Anti-inflammatory drugs	Purinergic P2X3 receptors antagonist	BAY1817080 (oral)	NCT04487431 NCT04471337 NCT04454424 NCT04423744	Phase I competed
	Monoclonal antibodies of IL-33	MT-2990 (intravenous)	NCT03840993	Phase II completed
	An IL-1 receptor antagonist	Anakinra (subcutaneous)	NCT03991520	Phase II recruiting

## Data Availability

Not applicable.

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
