# Peer review of "A Lifelong Impact on Endometriosis: Pathophysiology and Pharmacological Treatment"

_ijms, 2023, doi:10.3390/ijms24087503_

Round 1

Reviewer 1 Report

Dear Authors,

The review is within the scope of the Special Issue. It is comprehensive and well-structured. Nevertheless, a minor revision seems necessary to improve the overall quality before the acceptance for publication. Please find below the comments and suggestions.

Introduction

- Line 38: An emphasis should be that the diagnosis is often delayed.

- Line 41: Include might be a more suitable that provide.

- Line 42: Which kind of treatment did you refer to as limited and with side effects?

- Line 47: Pain does not seem to be in the scope with the other mentioned mechanisms.

- Line 49: Review should replace study.

The pathophysiology of endometriosis

- Line 52: The subtitle should also include the Mullerian remnants theory.

- Line 61: It would be more precise to state that the included theories do not completely explain the pathogenesis.

- Lines 91-2: Please make an additional interpretaion of the percentages. Did they actually represent the increase in the probability?

- Lines 99-100: The readership would benefit from the more detailed biological annotation of the quoted genes.

- Lines 107-9: The sentence introduces certain ambiguity regarding the usefulness of QWAS. Please consider rephrasing to avoid such an impression.

- Lines 115-6: The sentence should emphasize that promotor methylation leads to gene silencing.

- There is no overview of the mechanisms involving histone modifications.

- Lines 123-31: Certain examples of non-coding RNAs would improve this section.

- Line 142: The sentence should emphasize that the accumulation of estradiol is involved in the activation of the genes.

- Line 178: A more suitable term should replace innate mononuclear phagocyte systems.

- Lines 186-90: The section would benefit from a brief explanation of the phenotype of macrophages (M1 and M2). Also, an emphasis should be that the affected women have both physiological endometria, in which M1 predominates, and ectopic portion(s), with M2 predominance.

Clinical features of endometriosis and its lifelong impacts

- Line 223: A more suitable explanation might be that the severity of endometriosis can be determined after surgery.

- Line 224: A textual or table summary of the scoring system would be relevant to the readers.

- Lines 250-2: Please consider the additional efforts to include examples of the placenta abnormalities and the adverse neonatal outcomes.

- Line 256: A neonates-related characteristic seems to be missing.

- Lines 259-60: Reference(s) should support the sentence.

Pharmacologic therapies in current clinical practice

- The general improvement in this section should address the lack of dosing regimens. Also, surgical considerations should be relocated to another part of the manuscript, such as the section dealing with endometriosis extent assessment.

- Lines 286-8: Please accept dividing the text into two sentences to enhance understanding.

- Lines 288-9: The sentence should emphasize that the pharmacological treatment can be classified into two groups according to the targeted outcome,

- Please consider including a short structural explanation of GnRH antagonists.

- In the revised manuscript, do not include the text about danazol because it is no more part of the therapy.

- Lines 399-410: Please consider including a comprehensive explanation of which agents entered the clinical trials and in which phase.

- Lines 412-20: Are these agents efficient only as a supportive therapy?

Management of endometriosis-related infertility

- Lines 444-9: Reference(s) should support the text.

- Line 455: Please include combined after ovulation induction.

Conclusions

- Lines 480-3: Please make additional efforts to include the most important systemic and long-term impacts. What are the most relevant mechanisms pending acknowledgment in the pathogenesis of endometriosis?

- Line 491: What are the most promising novel targets?

Figures

- Please consider the additional effort to include more details in the legends. Furthermore, if schemes were modified from the literature, please add the original source to the reference list.

- For Figures 1 and 2, please emphasize the depicted mechanisms. The suggestion also refers to the corresponding sections in the text.

- In Figure 3, please indicate the references for the chemical structures.

Reviewer 2 Report

1. The included disease spectrum, which is related to endometriosis looks insufficient. There are multiple diseases that are known to be related lifelong. Considering this is a review paper, the range of mentioned disease spectrum isn't enough. 

2. There are a few references and studies, meta-analyses regarding EAOC. The contents needs to be modified thoroughly. 

3. The view regarding ART and pregnancy seems to be biased. The authors have to include either established recommendations or generally accepted expert opinions. Of course, if there are objective 'findings' supported by evidence, that will be the best of all.  

Reviewer 3 Report

The manuscript ‘ A lifelong impact in endometriosis: pathophysiology and 2 pharmacological treatment’ is well written, illustrated and documented.

A major problem is that pathophysiology is still exclusively viewed as 100 years ago with retrograde menstruation and implantation, and metaplasia. However, this cannot explain hereditary changes or clonal aspects or endometriosis in men, or heterogeneity of endometriosis lesions.  The key problems that should be addressed when discussing medical therapy are whether the endometriotic cell is endometrium (in an abnormal location) or whether the endometriosis has undergone genetic or epigenetic changes. Closely related to this are the many aspects associated with endometriosis: some are pre-existing to endometriosis others is a consequence: e.g; decreased NK cells and obstetrical changes do not change after excision of endometriosis suggesting that they are pre-existing.

To discuss hormonal therapy for endometriosis it would be useful to address normal steroid hormone concentrations in peritoneal fluid and concentrations after medical therapy.  

The text contains ‘may’ 19 times: for the modal reader it would be nice to differentiate between speculation and likely mechanisms.   

With all appreciation of the text please consider these comments as suggestions to help with our understanding of endometriosis. Reviewing is difficult and might introduce a personal bias, for which I apologise if happening.

Round 2

Reviewer 3 Report

Thank you for the revision.  However my 2 main concerns have not been addressed: Is the endometriosis cell a normal endometrial cell in an abnormal location or has the cell become different.  Second the high progesterone concentrations in peritoneal fluid have not been addressed when discussing medical therapy. 

However, I do appreciate the effort of writing this review, although it leaves me a feeling of a traditionally biased, selective view. 

To avoid a conflict because of personal opinions,  I therefore leave the decision to the responsible editor 
